# Association between nurses' knowledge and attitudes and caring behaviors toward people with opioid use disorder

Inyene Edem Essien-Aleksi[1]*, Yuan Zhang[2], Don Roosan[3], Tracie McPadden[4], Leslie Rideout[5], Michael Martin[5], Paula-Jo Beniers[5], Amy Lund[5], Danielle Leone-Sheehan[6], Alysse Wurcel[7]

**1** School of Nursing and Health Sciences, Merrimack College, North Andover, Massachusetts, United States of America, **2** Solomont School of Nursing, Zuckerberg College of Health Sciences, University of Massachusetts Lowell, Lowell, Massachusetts, United States of America, **3** School of Engineering and Computational Science, Merrimack College, North Andover, Massachusetts, United States of America, **4** Lowel General Hospital, Lowell, Massachusetts, United States of America, **5** Tufts Medical Center, Boston, Massachusetts, United States of America, **6** Northeastern University, Boston, Massachusetts, United States of America, **7** Boston University, Boston, Massachusetts, United States of America

☺ These authors contributed equally to this work.
* essienaleksi@merrimack.edu

## Abstract

### Background

Hospital nurses are crucial in inpatient OUD care, but knowledge gaps and negative attitudes may affect care quality. Few studies have examined the relationship between nurses' knowledge and attitudes and their caring behaviors toward patients with OUD in hospital settings.

### Method

We conducted a descriptive, exploratory cross-sectional study among hospital nurses at two northeast U.S hospitals. To explore nurses' perspectives, we used validated instruments to assess knowledge and attitudes [DDPPQ] and caring behaviors [CBI-6]. DDPPQ and CBI-6 scores were treated as ordinal composite measures and summarized using medians and interquartile ranges. Group differences were examined across demographic and work-related factors using Mann–Whitney U and Kruskal–Wallis tests. Associations between DDPPQ and CBI-6 scores were analyzed using Spearman's rank correlation with bootstrapped confidence intervals. Median (quantile) regression ($\tau = 0.50$) was used to assess adjusted associations.

### Results

A total of 125 nurses completed the survey. Most nurses were female (94%), white (82%), and employed full-time (87%). Higher DDPPQ scores, indicating lower knowledge and more negative attitudes, differed by years of experience and hospital site.

**Data availability statement:** All datasets can be accessed at Open ICPSR. Here is the URL to the dataset: https://www.openicpsr.org/openicpsr/project/238465/version/V1/view.

**Funding:** The author(s) received no specific funding for this work.

**Competing interests:** The authors have declared no competing interests.

CBI-6 scores varied by shift type, with permanent night-shift nurses reporting the highest median caring behaviors. Higher DDPPQ scores were significantly associated with lower caring behaviors ($\rho = -0.42$; 95% CI: $-0.52$ to $-0.30$; $p < 0.0001$). Even after adjusting for shift type, years of experience, and hospital site, higher DDPPQ score remained significantly associated with lower CBI-6 score ($\beta = -0.11$; 95% CI: $-0.16$ to $-0.06$).

## Conclusion

Hospital nurses with lower knowledge and more negative attitudes reported lower caring behaviors toward patients with OUD. These findings identify knowledge and attitudes as potential modifiable factors in inpatient OUD care and support future research to clarify causal pathways and explore the potential of educational and empathy-based strategies to improve OUD care.

## Introduction

Medications for opioid use disorder (MOUD) have transformed hospital-based care for opioid use disorder (OUD) [1], reducing overdose deaths and complications [2–5]. Each year, more than one million adults in the United States seek hospital-based care for OUD [6]. Yet many still face higher rates of post-hospitalization complications due to limited overdose prevention, delayed MOUD initiation, and inadequate follow-up after discharge [6,7]. Despite several interdisciplinary hospital-based partnerships to improve post-hospitalization outcomes for patients with OUD [6,8], major care gaps persist.

Nurses play a central role in addressing disparities in OUD care [6,9]. Representing approximately 23% of the U.S. healthcare workforce, nurses are uniquely positioned to implement and scale evidence-based interventions across healthcare settings [9–12]. For example, integrating trained nurse care managers into a primary care OUD treatment program has been shown to increase MOUD access and treatment uptake [9]. However, relatively few studies have examined interventions specifically targeting hospital nurses to improve care for patients with OUD. Existing research suggests that nurses may hold negative attitudes and misconceptions about evidence-based OUD treatments, particularly regarding the use of MOUD, which may influence caregiving practices [13]. In addition, nurses across diverse clinical settings report limited training and knowledge gaps, particularly regarding pain assessment and management in the context of OUD, as well as challenges in communicating patients' concerns to the broader care team [14]. These educational gaps, compounded by stigma surrounding OUD, may hinder nurses' ability to deliver high-quality, patient-centered OUD care [13,14].

Multiple factors—including education, clinical experience, burnout, and workplace context shape nursing practice and influence the quality of OUD care [9,13–15]. Because hospital nurses are often the first point of contact for patients with OUD, understanding the factors that influence their care practices is essential for quality

improvement initiatives aimed at improving inpatient OUD care. A multi-construct framework for evaluating health-related behaviors and educational interventions—encompassing knowledge, attitudes (or beliefs), and behaviors—has been widely applied across public health and educational research [16–19]. This framework provides a comprehensive approach to examining human behavior and educational outcomes [16]. Several nursing studies have applied these constructs to explore the relationships among knowledge, attitudes, and care practices across a range of health conditions [15,17]. However, a critical gap remains: no study has directly examined whether inpatient nurses' knowledge and attitudes toward OUD are associated with the care they provide to hospitalized patients with OUD. Addressing this gap may inform the development of targeted interventions to improve nursing care in inpatient OUD settings.

Within this framework, knowledge, according to Bloom's taxonomy, is the retrieval and understanding of previously learned information [20]. Attitudes are conscious or unconscious beliefs or perceptions that elicit emotional or psychological responses [16]. It has been conceptualized that nurses' professional and personal experiences, knowledge, and interactions within their environment shape their beliefs, attitudes (positive or negative), and actions related to care [21]. Caring behavior, as such, is an action-oriented practice that reflects interactions between care providers and recipients and contributes to patient outcomes [16]. Validated instruments such as the Care Behavior Index–6 (CBI-6) and the Drug and Drug Problems Perceptions Questionnaire (DDPPQ) have been widely used to assess nurses' caring behaviors, as well as their knowledge and attitudes toward patients across various health conditions [22–24]. Evidence from non-OUD contexts suggests that lower levels of knowledge and more negative attitudes among nurses are associated with poorer care delivery and adverse patient outcomes [25–27]. However, these relationships have not been adequately examined in the context of substance use disorder or OUD. As a result, limited evidence exists regarding patterns of hospital nurses' knowledge and attitudes toward OUD and how these factors relate to their caring behaviors. This exploratory descriptive study was designed to address this gap. Accordingly, this study was guided by the following research questions:

1. Are there significant differences in nurses' perceived knowledge and attitudes, and caring behaviors toward hospitalized adults with OUD across demographic and work-related characteristics in inpatient settings?

2. Are nurses' perceived knowledge and attitudes toward OUD significantly associated with their caring behaviors toward hospitalized adults with OUD in inpatient settings?

## Methods

### Study design

We conducted a descriptive-exploratory cross-sectional study at two adult inpatient hospitals: a nonprofit community and a teaching hospital in the northeast U.S. This study examined differences in nurses' knowledge and attitudes toward OUD, as well as their caring behaviors, across demographic and work-related characteristics. As a secondary aim, we explored the association between nurses' perceived knowledge and attitudes toward patients with OUD and their caring behaviors.

### Study sample

The target population included all adult inpatient nurses, excluding those in the emergency room and outpatient clinics. Study participants were recruited using convenience and purposive sampling methods. Nurses at the two hospital settings who self-identified as 18 years or older and worked full-time, part-time, or per diem in adult inpatient units were eligible to participate. Exclusion criteria included incomplete questionnaires, nurses not actively practicing at the bedside, travel nurses, nurse managers, administrators, and nurses with less than six months of inpatient experience. We excluded nurses with less than 6 months of experience to ensure adequate exposure to OUD-related clinical cases and sufficient knowledge development. Nurse administrators and non-bedside nurses were excluded because their roles do not involve direct patient care, and their knowledge and professional development may differ from those of bedside nurses. These

criteria aligned with the study's objectives and ensured inclusion of perspectives from direct care nurses. Sample size was calculated using G*Power 3.1 software, with a medium effect size (d = 0.50), 80% power, and a 0.05 significance level, yielding a required sample size of 84 participants.

## Data collection

Data were collected at the two Northeast hospitals between September 16 and December 10, 2024. Participants were recruited through flyers posted in selected inpatient units, nursing grand rounds, newsletters, and meetings of the nursing leadership committee. A HIPAA-compliant mobile application and email were used to recruit participants. Surveys were distributed electronically via REDCap. The first page of the online survey explained the study's purpose, procedure, potential benefits, and risks, and required participants to read and approve before proceeding to the survey questions. No identifying information was collected in the surveys. Participants were assured that their employer would not have access to their responses and that participation would not affect their employment status. Upon completing the survey, participants were entered into a raffle to win a $50 Amazon gift card. A total of 20 raffle prizes were awarded. To minimize selection and response bias, the winners of the raffle were selected at random via an independent electronic system not linked to the research team or their survey responses. Permission to conduct this study was obtained from the Institutional Review Boards (IRB) at Merrimack College (IRB -FY24-25-2) and the Hospital System (IRB #00004941).

## Study measures

**Dependent variable.** *Caring Behaviors:* Nurses' perception of caring behavior was assessed using the revised Caring Behavior Inventory-6 (CBI-6) scale. This scale was originally developed by Wolf and colleagues [24,28] and later revised by Coulombe et al. [23]. The CBI-6 assesses caring behaviors using 6 items rated on a 6-point Likert scale ranging from 1 ("never") to 6 ("always"). Item scores are summed to produce a total score ranging from 6 to 36, with higher scores reflecting higher perceived levels of nurse caring. The CBI-6 does not have clinically validated cut-off scores. Prior studies have reported good internal consistency for the CBI-6, with Cronbach's α value of 0.89 [23,29]. In the current sample, the CBI-6 demonstrated good internal consistency reliability (Cronbach's α = 0.82).

**Independent variable.** *Knowledge and Attitudes*: Nurses' perceived knowledge and attitudes toward people with OUD were measured using the Drug and Drug User Problems Perceptions Questionnaire (DDPPQ). The scale was originally developed by Cartwright [22] as the Alcohol and Alcohol Problems Perception Questionnaire (AAPPQ), a 30-item instrument assessing healthcare professionals' self-assessed knowledge and attitudes toward caring for people with alcohol-related problems. The AAPPQ was later revised and validated by Watson et al. [30], who renamed it the DDPPQ to reflect broader applicability to drug-related problems. The DDPPQ consists of 20 items rated on a 7-point Likert scale ranging from 1 ("strongly agree") to 7 ("strongly disagree"). Item scores are summed to produce a total DDPPQ score, with a theoretical range of 20–140. Lower DDPPQ scores indicate higher perceived knowledge and more positive attitudes toward care for people with drug-related problems, while higher DDPPQ scores indicate more negative perceptions and lower knowledge [30]. The DDPPQ scale includes five conceptual subscales: role adequacy, role support, role legitimacy, role-related self-efficacy, and job satisfaction. The DDPPQ lacks clinically validated cut-off scores for interpretation, and prior studies have used diverse analytical approaches, including total and subscale scores or grouping items into knowledge- and attitude-related domains [30,31]. In the current study, the DDPPQ composite score was calculated as the sum of all item responses and used as the outcome measure. Consistent with the literature [30,31], the internal consistency reliability of the DDPPQ in this sample was good (Cronbach's α = 0.88).

**Covariates.** Demographic variables included age (categorized into age groups), gender, race (White, Black, Asian, or other), and self-identified ethnicity (Hispanic vs. non-Hispanic). Work-related variables (factors) included primary nursing role (RN [bedside] vs. nurse educator), employment status (full-time, part-time, or per diem), highest nursing degree (associate degree, bachelor's degree, or master's degree), years of nursing experiences (≤ to 1 year, 1–5 years, 6–10

years, or over 15 years), shift type (permanent day, permanent evening, permanent night, and rotating shift), hospital site (university/teaching [urban]) vs. community hospital), and frequency of work-related exposure to patients with OUD. Variables significantly associated with DDPPQ and CBI-6 in bivariate analyses were subsequently adjusted for in the quantile regression models.

### Data analysis

Data analyses were conducted using SAS 9.4 (SAS Institute, Cary, NC) within a descriptive, exploratory framework. Internal consistency for the DDPPQ and CBI-6 scales was evaluated using Cronbach's alpha. Total DDPPQ and CBI-6 scores were derived from multiple Likert-type items and calculated as the sum of all item responses. These scores were treated as ordinal composite scores rather than continuous variables [31,32]. Thus, our analytical plan does not assume normality, interval-level measurement, or random assignment, and relies solely on nonparametric statistical methods [32]. Descriptive statistics summarized nurse demographic and work-related factors and were reported as frequencies and percentages (Table 1). Measures of central tendency for the ordinal DDPPQ and CBI-6 total scores were summarized using medians and interquartile ranges (IQRs), and the corresponding theoretical score ranges were also reported. We used rank-based nonparametric tests to examine differences in total DDPPQ and CBI-6 scores across nurse demographic and work-related factors [32]. The Mann-Whitney U test was used to compare total DDPPQ or CBI-6 scores between two groups, and the Kruskal-Wallis test was used for three or more groups (Tables 2 and 3). When the Kruskal-Wallis test was significant, post hoc pairwise comparisons were conducted descriptively using the Dwass–Steel–Critchlow–Fligner method.

Several subgroups, including race, highest nursing degree, employment status, and shifts, were too small to analyze. Therefore, we merged (collapsed) them into meaningful analytic groups. Decisions to merge these marginal subgroups were guided by the distribution of values within their categories and by conceptual similarity. Extremely small predefined subgroups, such as ethnicity and primary nurse role, were excluded from the bivariate analysis. Their sizes were insufficient for reliable interpretation. Given the exploratory nature of this study and the unequal subgroup sizes, we interpreted the post hoc results with caution.

We examined associations between nurses' knowledge and attitudes (DDPPQ) and caring behaviors (CBI-6) for patients with OUD using Spearman's rank correlation ($\rho$). We estimated 95% confidence intervals by a bootstrap procedure [33]. Median-quantile regression was used to estimate adjusted associations between DDPPQ and the median total CBI-6 score ($\tau = 0.50$). This approach is consistent with recommendations for ordinal data analysis [34,35]. We selected median-quantile regression to model caring behaviors because CBI-6 scores showed a negatively skewed distribution and lacked validated clinical cutoffs to guide meaningful stratification [35]. The regression model was adjusted for demographic and work-related factors significantly associated with DDPPQ and CBI-6 scores in bivariate analyses. These variables included shift type, years of nursing experience, and hospital site (Table 4, Panel B). The adequacy of the model was evaluated using statistical significance, confidence intervals, and consistency with non-parametric analyses. The regression coefficient represents the estimated change in the total CBI-6 median score for a one-unit increase in the DDPPQ score. All statistical tests were two-sided, with significance set at $p < 0.05$. Given the cross-sectional design and the use of ordinal scales, the findings should be interpreted as associative and exploratory rather than causal.

## Results

### Descriptive and bivariate analyses

A total of 224 questionnaires were distributed to prospective participants. Of these, 125 were fully completed. Ninety-nine questionnaires were excluded because of incomplete responses. Table 1 summarizes nurse demographic and work-related variables. Most participants were female (94%), white (82%), and employed full-time (87%). Additionally, 41% worked rotating nursing shifts. Nearly all participants (98%) were bedside or direct care nurses. Thirty-eight percent

**Table 1. Nurses' demographic and work-related variables (*n* = 125).**

| Variables | All Participants n *(%)* |
|---|---|
| **Gender** | |
| Female | 114 (94.21) |
| Male | 6 (4.96) |
| Prefer Not to Answer | 1 (0.83) |
| **Age-Group (*years*)** | |
| 20–25 | 23 (19.33) |
| 26–30 | 37 (31.09) |
| 31–40 | 28 (23.53) |
| 41–66 | 31 (26.05) |
| **Race** | |
| White | 103 (82.40) |
| Black | 2 (1.60) |
| Asian | 4 (3.20) |
| Other | 5 (4.00) |
| Prefer not to answer | 6 (4.80) |
| **Ethnicity*** | |
| Non-Hispanic | 116 (98.31) |
| Hispanic | 2 (1.69) |
| **Primary Nursing Role*** | |
| RN (bedside) | 119 (98.35) |
| Nurse Educator | 2 (1.65) |
| **Highest Nursing Degree** | |
| Associate degree | 8 (6.56) |
| Bachelor's degree | 97 (79.51) |
| Master's degree | 17 (13.93) |
| **Employment Status** | |
| Full-time | 106 (87.60) |
| Part-time | 11 (9.09) |
| Per-diem | 4 (3.31) |
| **Shift Type** | |
| Permanent day | 39 (31.71) |
| Permanent evening | 4 (3.25) |
| Permanent night | 29 (23.58) |
| Rotating | 51 (41.46) |
| **Years of Nursing Experience** | |
| Less than 1 year | 8 (6.50) |
| 1–5 years | 49 (39.84) |
| 6–10 years | 21 (17.07) |
| 11–15 years | 18 (14.63) |
| Greater than 15 years | 27 (21.95) |
| **Hospital Site** | |
| University/Teaching (city) Hospital | 74 (60.16) |
| Community Hospital | 49 (39.84) |

*(Continued)*

**Table 1.** (Continued)

| Variables | All Participants n *(%)* |
|---|---|
| **Frequency of Work-Related Exposure to Patients with OUD** | |
| Every day | 17 (13.82) |
| ≥1 day a week | 47 (38.21) |
| ≥1 day a month | 37 (30.08) |
| ≥1 day a year | 19 (15.45) |

**Note:** Values reported are all nurses' demographic and work-related variables for the sample. *n: number;* %: percentage.

cared for patients with OUD at least once a week. The median DDPPQ total score was 56.0 (IQR: 46.0–66.0), and the median CBI-6 total score was 31.0 (IQR: 28.0–34.0). Both measures showed distributions consistent with ordinal composite scores. CBI-6 scores clustered near the top of the possible (theoretical) range, suggesting a ceiling-skewed distribution.

**Differences in DDPPQ scores by nurses' demographic and work-related factors.** We used Kruskal-Wallis and Mann-Whitney U tests to examine differences in DDPPQ scores across nurses' demographic backgrounds and work-related factors (Table 2). No significant differences in scores were found across age groups, gender, race, educational level, employment status, shift type, or frequency of work-related exposure to patients with OUD. Median rank DDPPQ scores differed significantly by years of nursing experience, $\chi^2(3) = 7.95$, $p = 0.047$. Nurses scored differently by years of experience: the 6–10-year group scored lowest, whereas the 11–15-year and 15+year groups scored higher. Post hoc analysis using the Dwass–Steel–Critchlow–Fligner method showed no statistically significant differences across specific categories of years of work experience ($p > 0.05$). DDPPQ scores also differed by hospital site ($Z = 2.49$, $p = 0.013$). Nurses at the community hospital had higher median rank scores than those at the University/Teaching [urban] hospital. This suggests that nurses in University/Teaching hospitals reported greater knowledge of OUD and displayed more favorable attitudes toward patients with OUD than their counterparts in the community hospital.

**Differences in CBI-6 scores by nurses' demographic and work-related factors.** Kruskal-Wallis and Mann-Whitney U tests were used to determine whether CBI-6 total scores differed among nurses with different demographic backgrounds and work-related factors (Table 3). There were no significant differences in caring behavior scores across age groups, gender, race, educational level, employment status, years of nursing experience, hospital type, or frequency of caring for patients with OUD. But median rank CBI-6 scores differed significantly by shift type ($\chi^2 (2) = 6.99$, $p = 0.030$). Specifically, nurses working permanent night shifts reported the highest median ranks, followed by rotating-shift nurses; those working permanent day/evening shifts reported the lowest. Post hoc analysis showed a statistically significant difference between permanent day/evening versus permanent night shifts ($p = 0.027$), with permanent night-shift nurses scoring higher. No other groups showed significant differences ($p > 0.05$).

**Association between DDPPQ and caring behaviors.** Spearman's rank correlation was used to examine the unadjusted association between total DDPPQ median scores (nurses' knowledge and attitudes) and total CBI-6 median scores (caring behaviors) for care given to patients with OUD (Table 4, Panel A). Higher DDPPQ scores were significantly and inversely associated with CBI-6 scores ($\rho = -0.42$, $p < 0.0001$), indicating that lower knowledge and more negative attitudes were associated with lower caring behaviors. The association was robust to bootstrap resampling (10,000 iterations), with a bootstrapped Spearman correlation of $\rho = -0.42$ (95% CI: −0.53 to −0.30). Median ($\tau = 0.50$) quantile regression was used to model the median of CBI-6 by DDPPQ scores, adjusting for shift type, years of nursing experience, and hospital site (Table 4, Panel B). Higher DDPPQ scores remained significantly associated with lower total CBI-6 median scores ($\beta = -0.11$; 95% CI: −0.16 to −0.06). No covariates were associated with caring behaviors at the median level.

**Table 2. DDPPQ total scores by nurses' demographic and work-related variables (*n* = 125).**

| Variables | Group (*n*) | Median (IQR) | Test statistics | *p*-value |
|---|---|---|---|---|
| **Gender** | | | | |
| Female | 114 | 56.0 (46-67) | | |
| Male | 6 | 53.0 (29-59) | Z = −0.73 | 0.466 |
| **Age-Group (*years*)** | | | | |
| 20–25 | 23 | 52.0 (44-63) | | |
| 26–30 | 37 | 58.0 (49-70) | χ² = 0.65 | 0.885 |
| 31–40 | 28 | 56.0 (47-67) | | |
| 41–66 | 31 | 54.0 (45-72) | | |
| **Race (collapsed)** | | | | |
| White | 103 | 55.0 (46-66) | | |
| Non-white† | 11 | 57.0 (41-63) | Z = −0.10 | 0.915 |
| **Highest Nursing Degree** | | | | |
| Associate degree | 8 | 56.5 (42-67) | | |
| Bachelor's degree | 97 | 56.0 (46-67) | χ² = 0.94 | 0.954 |
| Master's degree | 17 | 56.0 (49-66) | | |
| **Employment Status (collapsed)** | | | | |
| Full-time | 106 | 56.0 (46-66) | | |
| Non-full-time† | 15 | 56.0 (45-72) | Z = 0.09 | 0.924 |
| **Shift Type** | | | | |
| Permanent day/evening | 43 | 59.0 (48-74) | | |
| Permanent night | 29 | 55.0 (41-61) | χ² = 4.33 | 0.110 |
| Rotating | 51 | 52.0 (45-64) | | |
| **Years of Nursing Experience (collapsed)** | | | | |
| ≤ 5 years † | 57 | 56.0 (50-64) | | |
| 6–10 years | 21 | 46.0 (32-58) | χ² = 7.95 | **0.047*** |
| 11–15 years | 18 | 56.0 (51-72) | | |
| Greater than 15 years | 27 | 55.0 (46-73) | | |
| **Hospital Site** | | | | |
| University/Teaching | 74 | 52.0 (45-61) | | |
| Community | 49 | 60.0 (51-72) | Z = 2.49 | **0.013*** |
| **Frequency of Work-Related Exposure to Patients with OUD** | | | | |
| Every day | 17 | 52.0 (44-66) | | |
| ≥ 1 day a week | 47 | 53.0 (41-63) | χ² = 1.73 | 0.784 |
| ≥ 1 day a month | 37 | 57.0 (46-68) | | |
| ≥ 1 day a year | 19 | 58.0 (50-64) | | |

**Note:** Values are reported as medians with interquartile ranges (IQR: Q1-Q3) for DDPPQ by nurses' demographic and work-related variables. Mann–Whitney U tests were used for two-group comparisons, and Kruskal–Wallis tests for comparisons involving three or more groups. † Cell combined for meaningful comparisons. Race: Non-white [Black, Asian, and other]; employment status: Non-full-time [part-time and per diem]; shift type: Permanent day/evening shifts; years of experience: ≤ 5 years [less than 1 year and 1–5 years]. DDPPQ = Drug and Drug Problems Perceptions Questionnaire; n = group sample size or number. *Bolded values are significant at p < 0.05.

## Discussion

We conducted a descriptive, exploratory study to examine hospital nurses' perceived knowledge and attitudes toward patients with opioid use disorder (OUD) and their caring behavior. We studied two adult inpatient settings. Using ordinal

**Table 3. CBI-6 total scores by nurses' demographic and work-related variables (*n* = 125).**

| Variables | Group (*n*) | Median (IQR) | Test statistics | *p*-value |
|---|---|---|---|---|
| **Gender** | | | | |
| Female | 114 | 31.0 (28-34) | | |
| Male | 6 | 31.5 (27-33) | Z = −0.42 | 0.672 |
| **Age-group (*years*)** | | | | |
| 20–25 | 23 | 32.0 (29-34) | | |
| 26–30 | 37 | 32.0 (28-35) | χ² = 3.08 | 0.378 |
| 31–40 | 28 | 30.0 (28-34) | | |
| 41–66 | 31 | 30.0 (28-34) | | |
| **Race (collapsed)** | | | | |
| White | 103 | 32.0 (28-34) | | |
| Non-White† | 11 | 30.0 (29-34) | Z = −0.12 | 0.900 |
| **Highest nursing degree** | | | | |
| Associate degree | 8 | 30.0 (29-34) | | |
| Bachelor's degree | 97 | 31.0 (28-34) | χ² = 0.62 | 0.732 |
| Master's degree | 17 | 31.0 (28-33) | | |
| **Employment status** | | | | |
| Full-time | 106 | 31.0 (28-34) | | |
| Non-full-time† | 11 | 30.0 (29-34) | Z = 0.03 | 0.974 |
| **Shift type (collapsed)** | | | | |
| Permanent day/evening† | 43 | 30.0 (26-33) | | |
| Permanent night | 29 | 33.0 (30-35) | χ² = 6.99 | **0.030*** |
| Rotating | 51 | 31.0 (28-34) | | |
| **Years of Nursing Experience (collapsed)** | | | | |
| ≤ 5 years† | 57 | 32.0 (30-34) | | |
| 6–10 years | 21 | 34.0 (29-35) | χ² = 6.68 | 0.082 |
| 11–15 years | 18 | 29.5 (25-32) | | |
| Greater than 15 years | 27 | 30.0 (28-34) | | |
| **Hospital Site** | | | | |
| University/Teaching | 74 | 32.0 (28-34) | Z = −0.55 | 0.578 |
| Community | 49 | 30.0 (28-34) | | |
| **Frequency of Work-Related Exposure to Patients with OUD** | | | | |
| Every day | 17 | 32.0 (29-35) | | |
| ≥ 1 day a week | 47 | 30.0 (28-34) | χ² = 1.43 | 0.837 |
| ≥ 1 day a month | 37 | 28.0 (28-34) | | |
| ≥ 1 day a year | 19 | 30.0 (27-34) | | |

**Note:** Values reported are medians with interquartile ranges (IQR: Q1-Q3) for total CBI-6 scores by demographic and work-related variables. Mann–Whitney U tests were used for two-group comparisons, and Kruskal–Wallis tests for comparisons involving three or more groups. † Cell combined for meaningful comparisons. Race: Non-white [Black, Asian, and other]; employment status: Non-full-time [part-time and per diem]; shift type: permanent day/evening shifts; years of experience: ≤ 5 years [less than 1 year and 1–5 years]. CBI-6: Care Behavior Index-6; n = group sample size or number; * bolded values are significant at *p* < 0.05.

and nonparametric analytic methods, we found that nurses overall reported higher levels of positive caring behaviors toward patients with OUD. This was evidenced by median total CBI-6 scores concentrated at the upper end of the scale. In contrast, knowledge and attitude scores were more variable. Across analyses, higher DDPPQ scores, which indicate lower perceived knowledge and more negative attitudes toward patients with OUD, were significantly inversely associated

**Table 4. Association between nurses' knowledge and attitudes (DDPPQ) and caring behaviors (CBI-6).**

**Panel A. Unadjusted association**

| Variables | n | Spearman's ρ | 95% Bootstrap CI | p-value |
|---|---|---|---|---|
| Total DDPPQ vs. total CBI-6 score | 125 | −0.42 | −0.52 to −0.30 | <0.0001 |

**Panel B.** Adjusted association between nurses' knowledge and attitudes (DDPPQ) and caring behaviors (CBI-6) with median-quantile regression [τ = 0.50]

| Predictor | n | Coefficient estimate (β) | 95% CI |
|---|---|---|---|
| **Total DDPPQ score** | 123 | **−0.11** | **−0.16 to −0.06** |
| **Shift type (collapsed)†** | | | |
| Permanent day/evening | 123 | −0.03 | −1.85 to 2.15 |
| Permanent night | 123 | 1.66 | −0.78 to 3.79 |
| Rotating | 123 | Reference | — |
| **Years of nursing experience (collapsed)\*** | | | |
| ≤5 years | 123 | 1.77 | −0.45 to 4.13 |
| 6–10 years | 123 | 0.89 | −2.66 to 3.36 |
| 11–15 years | 123 | −0.80 | −3.50 to 2.44 |
| >15 years | 123 | Reference | — |
| **Hospital Site§** | | | |
| University/Teaching hospital | 123 | −0.06 | −1.93 to 1.40 |
| Community hospital | 123 | Reference | — |

**Note:** Table 4 Panel A represents the unadjusted association between DDPPQ and CBI-6 scores using Spearman's rank-order correlation. Confidence intervals for Spearman's ρ were calculated using nonparametric bootstrap resampling (10,000 iterations). Table 4, Panel B reports the adjusted association between DDPPQ and CBI-6 scores from a median-quantile regression (τ = 0.50). τ represents the quantile of the outcome distribution modeled; τ = 0.50 corresponds to the median total CBI-6 scores. β coefficients represent the change in the median total CBI-6 score associated with a one-unit increase in the median total DDPPQ score. CI = 95% Confidence interval estimated using the inverse rank method. † Reference category for shift type: rotating shifts; * reference category for years of nursing experience: >15 years; § reference category for hospital site: community hospital. n = number. Higher DDPPQ scores indicate more negative attitudes and lower knowledge; higher CBI-6 scores indicate greater caring behaviors.

with lower caring behavior scores. This association was assessed using Spearman's rank-order correlation with boot-strapped confidence intervals. It also remained significant in adjusted median-quantile regression analyses. In median regression, we adjusted for shift type, years of experience, and hospital site. The consistency of findings across analyses and after adjustment strengthens confidence in the observed association between knowledge and attitudes and nurse caring. Median regression was selected to reduce the sensitivity of total CBI-6 scores to ceiling-skewed effects [35]. This approach also suited exploratory analyses of ordinal composite scores, especially in the absence of clinically validated cut-off scores for meaningful categorization of CBI-6 scores. The cross-sectional study design precludes causal inference; therefore, we only claim an association between the examined variables.

At the bivariate level, we found no significant differences in knowledge and attitudes across age groups, gender, race, educational level, employment status, shift type, or frequency of work-related exposure to patients with OUD. Nurses' knowledge and attitudes differed only by years of experience and hospital type (teaching/university [urban] vs. community hospital). Interesting, nurses with 6–10 years of experience tended to report more positive attitudes and knowledge than those with more than 11 years of experience. However, post hoc comparisons showed no significant differences among the categories of years of nursing experience. Contrary to our expectations, we expected nurses with more years of practical experience to have greater knowledge and more favorable attitudes toward OUD; however, our sample suggested otherwise. An observational study exploring nurses' self-assessed attitudes and knowledge about substance use disorder across hospital settings in the United States found that older nurses with more years of experience tended to exhibit greater knowledge and more positive attitudes than their younger counterparts [36]. In contrast, another study

of emergency room nurses found that younger nurses and those with less experience had more positive attitudes and were more willing to screen for and promote evidence-based OUD care, including harm reduction strategies and MOUD (buprenorphine) initiation, than their older, more experienced colleagues [15]. But another study found no differences in ED nurses' attitudes, knowledge, and comfort in caring for patients with OUD by their years of experience [37]. The observed discrepancies in findings may be attributable to the predominant use of observational designs in substance use disorder-related nursing research, which often rely on self-reported instruments that are subject to measurement bias [15,36–38]. Furthermore, the current nursing educational programs have become increasingly diverse, encompassing multiple entry options, such as traditional BSN, two-year RN-BSN, accelerated, or direct-entry master's programs for non-nurses, making age and years of experience (often linked to age) unreliable indicators of nurses' knowledge or clinical preparedness [39,40]. As such, the literature remains mixed regarding the influence of age and experience on perceptions of nursing knowledge and attitudes in clinical practice [13].

We also found overall and between-group differences in nurses' knowledge and attitudes by hospital site. Nurses working in the community hospital tended to report lower knowledge and more negative attitudes than those in the teaching or university [urban] hospital. This finding was expected, as it has been suggested that nurses in resource-rich environments, such as university or teaching hospitals, may benefit from exposure to higher-acuity patients [41]. These nurses may also have greater access to OUD-specific education and interdisciplinary support for OUD care [41,42]. Overall, these findings suggest that both individual-level factors, such as experience and organizational context, may influence aspects of nurses' perceptions (attitudes) and knowledge related to OUD care.

Finally, this study explored differences in nurses' caring behaviors across demographic and work-related factors. There was no significant difference in nursing caring behavior across age groups, gender, race, educational level, employment status, or frequency of work-related exposure to patients with OUD. We had hoped to explore whether caring behaviors differed between direct and non-directed nurse roles. We expected bedside nurses to demonstrate more positive caring practices, as supported by previous studies [13,42]. However, given the significantly low cell size in this subgroup, this analysis was not conducted. We found significant differences only among shift types. Specifically, permanent night-shift nurses tended to report higher caring scores compared to day/evening or rotating-shift nurses. Although the reasons for this finding cannot be determined from the current data, possible explanations include differences in staffing patterns, patient–nurse interactions, workload patterns, or team dynamics across shifts [42–45]. Prior literature has demonstrated that night-shift nurses often manage similar complex patient profiles with greater autonomy and fewer competing demands, which may influence perceptions of relational care [46]. Other factors, not well explored in the literature and current data, such as stigma, burnout, compassion fatigue, and differing education and training pathways, may also influence caring behaviors toward people with substance use disorder [14,36,38,42,47–49]. Therefore, this finding warrants further studies examining contextual and workflow factors across shifts and units on nursing caring behaviors.

Overall, the study's findings underscore the importance of nurses' knowledge and attitudes as potential modifiable factors influencing caring behaviors toward patients with OUD in inpatient settings [36,37,49,50]. While this study was descriptive and exploratory, the results align with prior literature demonstrating an association between negative attitudes and knowledge and less supportive care practices among healthcare providers [51]. Notably, relatively few studies have examined how nurses' knowledge and attitudes relate directly to caring behaviors in hospital-based OUD care. Only Kratovil et al. [36] documented low levels of knowledge and negative attitudes among inpatient nurses caring for patients with substance use disorder, but did not examine how these factors relate to care behaviors. To our knowledge, this study is among the first to demonstrate a direct association between knowledge and attitudes and caring behaviors among inpatient nurses caring for patients with OUD. Even in that view, we still want to emphasize that these findings reflect associative relationships rather than causal effects. But the robustness of the association across different analytic methods supports the premise that nursing should evaluate whether certain educational and organizational strategies targeting improvements in OUD-related knowledge and attitudes may improve inpatient care experiences and outcomes

[36,50–52]. Given persistent disparities in hospital-based OUD care and the proven effectiveness of substance use–related training programs, future research should explore causal links between knowledge and attitudes, and caring behaviors. These results could support efforts to develop evidence-based interventions to address nurses' knowledge and attitude gaps in OUD care [36,51].

We acknowledge important limitations of this study. First, we used a descriptive, exploratory study design, drawing on a sample from two hospitals within a single integrated healthcare system. As such, our findings show only observational associations and do not imply causal relationships between knowledge and attitudes and caring behaviors. Second, the generalizability of the findings to other inpatient settings is limited. The study relied on convenience and purposive sampling methods, which introduces potential selection bias. For example, our sample was predominantly female (98%). Male nurses were underrepresented in this study, since approximately 12% of the national nursing workforce is male [10]. In several substance use studies, female nurse participants were overrepresented [9,14,36]. The reason for this issue is unclear; however, this further limits the generalizability of the study findings to a broader population. The study had an attrition rate of 44.2% (125/224). We restricted the analytic sample to complete responses to avoid missing data. However, this may have introduced selection bias and threatened internal validity. We excluded nurse administrators from the study sample, which may have led to missed insights from managers and administrators on nursing OUD care. We explored several strategies to mitigate potential sampling biases in the study. For example, we distributed the survey via a HIPAA-compliant app. Recruitment was internal, and participation was voluntary. Incentives may have increased participation. To minimize self-selection and response bias, raffle winners were randomly selected via an electronic draw. We used multiple recruitment methods (e.g., email, flyers, mobile app, meetings, and in-person) to minimize recruitment bias and to achieve a diverse sample. Lastly, there are several methodological and analytical limitations. We relied on self-reported measures, albeit using well-validated instruments to assess knowledge and attitudes and caring behaviors. These self-reported measures may be subject to response, recall, and social desirability biases, especially for caring behaviors. In addition, DDPPQ and CBI-6 scores were derived from Likert-type items and were therefore treated as ordinal composite measures. Analyses examining differences across demographic and work-related characteristics were further limited by a substantial imbalance in the distribution of subgroup categories. For example, predefined subgroups with small counts were excluded from group-difference comparisons. Certain categories were merged (collapsed) based on their distributions and conceptual similarity, enabling meaningful subgroup analyses. Although we used appropriate nonparametric methods, these differences may not fully capture the complexity and heterogeneity of nurses' knowledge and attitudes and caring behaviors in OUD care. Therefore, the reported bivariate results should be interpreted with caution.

Despite the noted limitations, this study contributes to the limited body of nursing research on OUD by advancing inquiry into this important area of nursing science. Future studies should recruit a larger, randomly selected, and more diverse nursing workforce, including entry-level nurses, male nurses, non-bedside nurses, individuals from diverse racial and ethnic backgrounds, varied employment classifications, shift types, and differing years of experience. These modifications would strengthen methodological rigor, address current analytical limitations, and enable a more robust examination of the research questions.

## Conclusions

In this descriptive and exploratory study, hospital nurses with lower knowledge and more negative attitudes tended to exhibit lower caring behaviors toward adult patients with OUD. Although these results were non-causal, they highlight that nurses' knowledge and attitudes are important modifiable factors in inpatient OUD care. Addressing educational gaps and reducing negative attitudes, beliefs, and stigma are central to improving hospital-based OUD care [4,36,48,53,54]. We recommend that future research employ longitudinal designs with larger, randomly selected samples to further examine how changes in nurses' knowledge and attitudes may predict changes in caring behaviors over time. Furthermore,

educational programs emphasizing humanistic and empathy-based approaches to nursing care should be explored to support nurses and strengthen their caring behaviors for patients with OUD [50,52].

## Acknowledgments

We thank the researchers and community partners at Merrimack College, Tufts Medical Center, as well as our research collaborators at the University of Massachusetts Lowell and Boston Medical Center.

## Author contributions

**Conceptualization:** Inyene Edem Essien-Aleksi, Yuan Zhang.

**Data curation:** Inyene Edem Essien-Aleksi, Yuan Zhang, Danielle Leone-Sheehan.

**Formal analysis:** Inyene Edem Essien-Aleksi.

**Funding acquisition:** Inyene Edem Essien-Aleksi.

**Investigation:** Inyene Edem Essien-Aleksi, Yuan Zhang, Danielle Leone-Sheehan.

**Methodology:** Inyene Edem Essien-Aleksi, Yuan Zhang, Danielle Leone-Sheehan.

**Project administration:** Inyene Edem Essien-Aleksi, Yuan Zhang, Tracie Mcpadden, Leslie Rideout, Michael Martin, Paula-Jo Beniers, Amy Lund, Danielle Leone-Sheehan.

**Resources:** Inyene Edem Essien-Aleksi.

**Software:** Inyene Edem Essien-Aleksi.

**Supervision:** Yuan Zhang.

**Validation:** Inyene Edem Essien-Aleksi, Yuan Zhang, Danielle Leone-Sheehan.

**Visualization:** Inyene Edem Essien-Aleksi.

**Writing – original draft:** Inyene Edem Essien-Aleksi.

**Writing – review & editing:** Inyene Edem Essien-Aleksi, Yuan Zhang, Don Roosan, Leslie Rideout, Danielle Leone-Sheehan, Alysse Wurcel.

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
