## [Decision Letter · Decision Letter 0]

29 Jul 2025

Influence of Nurse’s Knowledge and Attitudes on Caring Behaviors for People with Opioid Use Disorder

PLOS ONE

Dear Dr.  Essien-Aleksi,

Thank you for submitting your manuscript to PLOS ONE. After careful consideration, we feel that it has merit but does not fully meet PLOS ONE’s publication criteria as it currently stands. Therefore, we invite you to submit a revised version of the manuscript that addresses the points raised during the review process.

Please address reviewers’ comments.

We look forward to receiving your revised manuscript.

Kind regards,

Majed Sulaiman Alamri, PhD

Academic Editor

PLOS ONE

Journal Requirements:

6. We note you have included a table to which you do not refer in the text of your manuscript. Please ensure that you refer to Table 4 in your text; if accepted, production will need this reference to link the reader to the Table.

Reviewers' comments:

Reviewer's Responses to Questions

**Comments to the Author**

1. Is the manuscript technically sound, and do the data support the conclusions?

Reviewer #1: Partly

Reviewer #2: Yes

Reviewer #3: Partly

Reviewer #4: No

Reviewer #5: No

2. Has the statistical analysis been performed appropriately and rigorously?

Reviewer #1: Yes

Reviewer #2: Yes

Reviewer #3: I Don't Know

Reviewer #4: No

Reviewer #5: No

3. Have the authors made all data underlying the findings in their manuscript fully available?

Reviewer #1: No

Reviewer #2: Yes

Reviewer #3: Yes

Reviewer #4: No

Reviewer #5: Yes

4. Is the manuscript presented in an intelligible fashion and written in standard English?

Reviewer #1: No

Reviewer #2: Yes

Reviewer #3: Yes

Reviewer #4: Yes

Reviewer #5: Yes

Reviewer #1: First of all, the topic is very interesting and distinctive and deserves recognition. However, scientific writing contains many mistakes. There are a lot of repetitions in the abstract part, and it is very long. There are many incorrect references in the introduction, and the references are not numbered correctly and do not start with number one. The method section lacks clarity. To make the results more clear, the tables need to be accompanied by commentary. Furthermore, there is no explanation as to the data availability.

Reviewer #2: The manuscript is clearly written and facilitates reader comprehension. Nevertheless, the authors are encouraged to use and refer to a standardized checklist to further improve the quality and transparency of the methodology section.

Reviewer #3: This manuscript addresses a highly relevant clinical and public health topic: the impact of nurses’ knowledge and attitudes on their caring behaviors toward patients with opioid use disorder (OUD). The study employs validated instruments (DDPPQ and CBI-6) and appropriate statistical methods. However, some methodological and reporting weaknesses must be addressed to strengthen the study’s rigor, reproducibility, and generalizability.

Participants and Sample

Observation: Inclusion/exclusion criteria are described, but no justification or power calculation for sample size is provided. Recommendation: Include a rationale or power analysis for the sample size, or acknowledge this as a limitation.

Bias

Observation: There is no discussion of potential selection bias, self-selection bias, or social desirability bias due to self-reporting and voluntary participation. Recommendation: Address these biases in the limitations section and suggest mitigation strategies for future research (e.g., random sampling, cross-validation, stronger anonymity).

Measurement and Analysis

Observation: The use of validated instruments and appropriate statistical techniques is commendable. However, as all measures are self-reported, there is risk of bias. Recommendation: Discuss the potential for response bias (e.g., overreporting of caring behaviors) due to the self-reported nature of both primary instruments.

Causal Inference

Observation: Some wording in the discussion may suggest causal relationships. Recommendation: Reinforce that, given the cross-sectional design, only associations—not causal relationships—can be inferred.

Reviewer #4: Abstract

The abstract attempts to summarize a dense and multifaceted study but becomes cluttered with too much background. The research objective gets somewhat buried. The phrasing “Additionally, we further examined differences...” is redundant.

Reporting results with specific statistics (β = -0.11, p < 0.0001) in the abstract is good, but the implications of these findings are not adequately synthesized. There’s minimal interpretation beyond stating the association.

The phrase “Nurses’ knowledge and attitudes play a crucial role...” is repeated in the abstract and again in the conclusion with near-identical wording, suggesting a lack of synthesis.

Introduction

The introduction excessively relies on citations (e.g., multiple references in one sentence) without critically integrating them. Some references seem included for volume rather than relevance.

Although the issue is important, the introduction lacks a clear conceptual model or hypothesis linking knowledge, attitudes, and caring behavior.

The gap in the literature is stated but not well substantiated. The claim that this is the “first study” to link these constructs lacks sufficient justification or contrast with existing studies.

Terms like “caring behavior” and “attitudes” are introduced without conceptual definitions, relying instead on the instruments (CBI-6, DDPPQ) to carry the burden of meaning.

Methods

The use of convenience and purposive sampling is a significant limitation, but this is not acknowledged until the discussion. This undermines generalizability and introduces bias.

The exclusion of nurse managers and educators is reasonable but not well-justified. Given their role in shaping attitudes and policy, their exclusion may miss important systemic insights.

The explanation of the DDPPQ scoring is confusing. The creation of stratified ranges (e.g., “very low,” “low,” etc.) seems arbitrary without validation or citation.

The methods mention the use of REDCap and a raffle incentive, but there's no reflection on how these may introduce self-selection or response bias.

No justification for the sample size is provided, which weakens confidence in the representativeness and power of statistical analyses.

Results

The section is filled with statistical comparisons but lacks interpretation. Reporting p-values and CI is appropriate, but there’s minimal explanation of what these mean practically.

The categorization of DDPPQ and CBI-6 scores into subjective ranges (e.g., high, moderate, low) is not grounded in prior literature, reducing replicability.

The volume and complexity of the tables may overwhelm the reader. Many differences are noted as statistically insignificant, suggesting a risk of fishing for significance.

The emphasis on statistical significance ignores potential Type I errors due to multiple comparisons without correction.

Discussion

The discussion makes strong claims about causality and implications despite the cross-sectional design. Phrases like “nurses’ knowledge and attitudes were the strongest predictors” suggest a causal relationship unsupported by the study design.

Many points are reiterated from earlier sections (e.g., “nurses in community hospitals had less favorable attitudes”), which inflates word count without adding insight.

Although some literature is cited, there’s a lack of critical integration. The findings are not well-situated within broader theoretical or empirical contexts.

Statements about how institutional culture or training might affect outcomes are speculative and unsupported by data from the study.

The limitations section is undercut by generic statements. The gender imbalance and response bias deserve deeper examination.

Conclusion and Implications

The conclusion pushes the policy relevance too hard without sufficient data to support its sweeping claims (e.g., EMR-integrated interventions).

Recommendations for educational interventions are vague. What kind of intervention? Delivered how? By whom?

The conclusion asserts that RN licensure status “contributes” to caring behavior, yet the earlier analysis shows only a weak association. The term “contributes” overstates the evidence.

Ethical and Data Transparency Statements

The ethics section is sufficiently detailed but fails to address potential coercion or social desirability bias given that recruitment was conducted internally and involved monetary incentives.

While the link to OpenICPSR is appreciated, the manuscript does not indicate whether de-identified datasets were reviewed for completeness or whether codebooks are available.

References

Several references are dissertations or lesser-known journals, which may weaken the scholarly weight. Peer-reviewed, high-impact studies should be prioritized.

Some studies are cited multiple times in different sections with near-identical phrasing, indicating potential padding.

Reviewer #5: 1. Title vs. Study Design

There is a fundamental inconsistency between the study title and the actual research design. The title implies a causal relationship (“influence”), whereas the study employs a cross-sectional, observational design using correlational analyses. Causal inference is not warranted in such a design.

Recommendation: Consider revising the title to reflect the observational nature of the study. For example:

“Association Between Nurses’ Knowledge and Attitudes and Caring Behaviors Toward People With Opioid Use Disorder”, or

“Exploring the Relationship Between Nurses’ Knowledge and Attitudes and Their Caring Behaviors...”

2. Clarity and Alignment of Objectives

The study's aim is not clearly and explicitly stated. The main objective appears to be to examine the relationship between nurses’ knowledge/attitudes and caring behaviors, while a secondary objective is to explore differences based on demographic and workplace factors. However, the latter is distorted by an erroneous analysis plan.

3. Study Design and Sampling

The study is described as cross-sectional, which is appropriate for examining associations. However, the wording in other parts of the manuscript (e.g., title and conclusions) suggests causal inference. This discrepancy should be addressed throughout to avoid overinterpretation of findings.

The use of convenience and purposive sampling introduces potential bias, especially regarding representativeness. While this is common in exploratory nursing studies, it limits generalizability and should be acknowledged.

Recommendation: Clarify whether a power analysis or sample size estimation was conducted a priori. If not, provide a rationale for the achieved sample size and its implications for statistical power.

3. Measurement of Key Constructs

a. Caring Behaviors (Dependent Variable)

The use of the Caring Behavior Inventory-6 (CBI-6) is appropriate and supported by internal consistency data. However, the categorization of CBI-6 scores into “low, moderate, and high” levels appears arbitrary, as the instrument lacks validated cut-off points.Correct the range of Cronbach's alpha of the CBI-6

Recommendation: Justify the rationale for the chosen score ranges, or alternatively, consider treating the CBI-6 score as a continuous variable in line with existing literature.

b. Knowledge and Attitudes (Independent Variable)

The DDPPQ is used as a combined measure of nurses’ "knowledge and attitudes." While this usage is consistent with some prior studies, the DDPPQ is more accurately described as a measure of perceived therapeutic competence and attitudes, not direct knowledge.

The stratification into “very low” to “high” levels based on Likert scale anchors is methodologically problematic, as DDPPQ has no validated thresholds and its interpretation is context sensitive.

Recommendation: Clearly distinguish between attitudes, role adequacy, and knowledge, as they are distinct theoretical constructs. If using the total DDPPQ score, provide a strong conceptual rationale and avoid over-interpreting results in terms of factual knowledge.

4. Recruitment and Ethical Procedures

Recruitment strategies and informed consent procedures are clearly described. The inclusion of a HIPAA-compliant recruitment platform and IRB approvals enhance the ethical credibility of the study.

Recommendation: It may be helpful to briefly comment on how the use of multiple recruitment methods may have influenced sample composition (e.g., risk of self-selection bias).

5. Data analysis plan.Although the manuscript reports p-values, these should be interpreted solely as descriptive indicators within the context of the study's design. The use of t-tests and one-way ANOVA implies assumptions of normality in the distribution of the response variable—assumptions that are not justified in this case. The variables analyzed are based on composite scores from Likert-type items, which do not warrant the application of parametric methods. Non-parametric approaches would be more appropriate for this type of data.

In addition, the study lacks an experimental or quasi-experimental design that would allow for valid group comparisons (e.g., by sex, ethnicity, or type of hospital). Since these group characteristics are not randomly assigned, the comparisons presented in the manuscript are not supported by the design and cannot be interpreted as causal.

Therefore, this manuscript should be considered as presenting preliminary descriptive findings. To support hypothesis testing and group comparisons, a follow-up study with a design that allows for such analyses would be necessary. For instance, an experimental design could include an intervention such as standardized training for healthcare personnel—an element not present in the current study.

Ensure the analytical strategy is fully transparent, including model types and rationale for covariate adjustment.

Clarify the interpretive boundaries of the design, particularly in relation to the causal language used elsewhere in the manuscript.

6. Results

The final analytic sample represents approximately 56% completion (125/224), which raises concerns about response bias. No information is provided about differences between completers and non-completers, which could affect the generalizability of findings.

Main Association: Knowledge/Attitudes and Caring Behaviors: The study reports a significant negative association between DDPPQ scores and CBI-6 scores, in both unadjusted and adjusted models. The use of simple linear regression (Model 1) followed by a generalized linear model (Model 2) with covariates may be methodologically acceptable. However, the term “Generalized Linear Model” is too broad and lacks specificity (e.g., link function, distribution). It is also unclear whether the assumptions of linearity, normality, and homoscedasticity were tested.The reported β coefficients are statistically significant but small in magnitude (β = -0.11 to -0.12), which should temper interpretations of practical significance.

DDPPQ and CBI-6 were stratified into categories (e.g., low/moderate/high), despite the lack of validated cut-offs for either scale. This raises concerns about data-driven categorization and the risk of artificial thresholds, which may reduce statistical power and obscure true relationships.

The finding that RNs scored higher on CBI-6 than nurse educators is statistically significant but may reflect differences in direct patient care exposure rather than intrinsic caring capacity.

The conclusion that holding an RN license contributes to caring behaviors may be an overinterpretation, especially given the small group size and possible confounding by role responsibilities.

The results section presents statistically significant and potentially meaningful findings. However, several issues require attention: Clarity in model specification and assumptions, Justification for categorization of continuous measures, Cautious interpretation of subgroup findings and their implications, Avoidance of causal language in describing observational associations.

7. Discussion

Add a paragraph analyzing possible reasons why experience and age did not show an association with attitudes and knowledge, contrasting with prior literature. For example, cultural differences, generational changes in training, or sample bias could be considered.

Elaborate more on the potential influence of professional role on caring behaviors, taking into account direct patient contact, workload, or level of ongoing professional development.

Include a comment on the gender representativeness of the sample relative to the national or regional nursing workforce demographics, which would help clarify the extent of this limitation.

Consolidate repetitive ideas to improve cohesion and avoid the reader perceiving redundancy.

Propose clear directions for future research, ideally focusing on how to overcome current limitations and deepen understanding of the factors shaping attitudes and behaviors related to OUD care.

**Do you want your identity to be public for this peer review?** For information about this choice, including consent withdrawal, please see our Privacy Policy

Reviewer #1: **Yes:** Ola Mousa

Reviewer #2: **Yes:** Peñarrubia-San-Florencio L

Reviewer #3: No

Reviewer #4: **Yes:** I Gede Juanamasta

Reviewer #5: **Yes:** Katya Cuadros-Carlesi

---

## [Author Response · Author response to Decision Letter 1]

29 Sep 2025

Manuscript Number: PONE-D-25-26604

Title: Association Between Nurses’ Knowledge and Attitudes and Caring Behaviors Toward People with Opioid Use Disorder

Dear Editor and Reviewers,

We sincerely appreciate your valuable comments. We have reviewed the suggestions from reviewers 1-5 and updated the manuscript accordingly. Please see our responses below in bold and in the manuscript for more details.

Sincerely,

Inyene Essien-Aleksi, PhD.

Response to the Editor

https://journals.plos.org/plosone/s/file?id=wjVg/PLOSOne_formatting_sample_main_body.pdfand
https://journals.plos.org/plosone/s/file?id=ba62/PLOSOne_formatting_sample_title_authors_affiliations.pdf]

We have formatted the manuscript to ensure it meets PLOS ONE’s style requirements.

We have revised the manuscript to ensure that no funding information is included anywhere. We received no funding from federal agencies or private organizations for this project.

We note that the grant information you provided in the ‘Funding Information’ and ‘Financial Disclosure’ sections do not match. When you resubmit, please ensure that you provide the correct grant numbers for the awards you received for your study in the ‘Funding Information’ section.

We have updated the Funding Statement and Financial Disclosure section to state that we received no funding from federal agencies or private organizations for this project.

Thank you for your comment. All relevant study data (de-identified) are now freely accessible. The data have been uploaded into the openICPSR data repository, with the link provided below. https://www.openicpsr.org/openicpsr/project/238465/version/V1/view

Please include your full ethics statement in the ‘Methods’ section of your manuscript file. In your statement, please include the full name of the IRB or ethics committee who approved or waived your study, as well as whether or not you obtained informed written or verbal consent. If consent was waived for your study, please include this information in your statement as well.

The method section has been revised to include the following information: “Permission to conduct this study was obtained from the Institutional Review Board (IRB) of Merrimack College (IRB -FY24-25-2) and Tufts Medicine (IRB #00004941). To collect data, the research team obtained electronic written informed consent from the participant prior to recruitment and data collection .”

We note you have included a table to which you do not refer in the text of your manuscript. Please ensure that you refer to Table 4 in your text; if accepted, production will need this reference to link the reader to the Table.

Thank you. We have revised the manuscript to include a reference to Table 4 in the Results section.

Thank you for the clarification. No reviewer recommended citing specific previously published works. However, we reviewed the manuscript's in-text citations and ensured the most relevant references are included to strengthen it.

Response to the Editor and Reviewer(s)

1. Reviewer #1: First of all, the topic is very interesting and distinctive and deserves recognition. However, scientific writing contains many mistakes. There are a lot of repetitions in the abstract part, and it is very long. There are many incorrect references in the introduction, and the references are not numbered correctly and do not start with number one. The method section lacks clarity. To make the results more clear, the tables need to be accompanied by commentary. Furthermore, there is no explanation as to the data availability.

Thank you, Reviewer #1, for your thoughtful and constructive feedback. We have made the following revisions to the manuscript: 1) Abstract: Significantly revised to eliminate repetition and shortened to meet PLOS ONE publishing guidelines. 2) Introduction: Corrected citation errors and ensured the references are numbered sequentially, starting at 1. 3) Methods: Revised for better clarity. 4) Results: Updated tables with additional commentary to enhance interpretation and clarity. 5) Data Availability: We have included a link to the openICPSR data repository in both the Supporting Information (within the manuscript) and the online submission form.

2. Reviewer #2: The manuscript is clearly written and facilitates reader comprehension. Nevertheless, the authors are encouraged to use and refer to a standardized checklist to further improve the quality and transparency of the methodology section.

Thank you, Reviewer #2, for your feedback. We have made revisions to improve the quality of the method section following the criteria outlined in PLOS ONE’s standardized checklist.

3. Reviewer #3: This manuscript addresses a highly relevant clinical and public health topic: the impact of nurses’ knowledge and attitudes on their caring behaviors toward patients with opioid use disorder (OUD). The study employs validated instruments (DDPPQ and CBI-6) and appropriate statistical methods. However, some methodological and reporting weaknesses must be addressed to strengthen the study’s rigor, reproducibility, and generalizability.

Participants and Sample

Observation: Inclusion/exclusion criteria are described, but no justification or power calculation for sample size is provided. Recommendation: Include a rationale or power analysis for the sample size, or acknowledge this as a limitation.

Thank you, Reviewer #3, for all your comments and constructive feedback. We have revised the data analysis paragraph to specify that power analysis was performed using G*Power and that a total of 84 participants were needed to detect a medium effect size (d = 0.50) with 80% power at a 0.05 significance level.

Bias Observation: There is no discussion of potential selection bias, self-selection bias, or social desirability bias due to self-reporting and voluntary participation. Recommendation: Address these biases in the limitations section and suggest mitigation strategies for future research (e.g., random sampling, cross-validation, stronger anonymity).

We agree with Reviewer #3 concerning the potential for biases. In the limitations section, we have now emphasized the potential for response bias, self-selection bias, and social desirability bias, which are inherent to the self-reporting nature of the instruments used in the study and the voluntary participation. We have suggested mitigation strategies for future studies, including random sampling, as well as cross-validation in a multi-site research framework to protect participants’ confidentiality/anonymity.

Measurement and Analysis

Observation: The use of validated instruments and appropriate statistical techniques is commendable. However, as all measures are self-reported, there is a risk of bias. Recommendation: Discuss the potential for response bias (e.g., overreporting of caring behaviors) due to the self-reported nature of both primary instruments.

We agree with Reviewer #3 about the risk of response bias. We highlighted this risk due to the self-reporting measures and described strategies to mitigate this limitation in the discussion section.

Causal Inference

Observation: Some wording in the discussion may suggest causal relationships. Recommendation: Reinforce that, given the cross-sectional design, only associations—not causal relationships—can be inferred.

We agree with Reviewer #3 for your comments regarding causal inference. We have thoroughly revised the discussion, limitations, and conclusion sections to include more specific language, indicating that our findings reflect only associations and not causal relationships. We have also changed the study title from “ Influence” to “ Association Between Nurses’ Knowledge and Attitudes and Caring Behaviors Toward People With Opioid Use Disorder”.

Reviewer #4: Abstract

The abstract attempts to summarize a dense and multifaceted study but becomes cluttered with too much background. The research objective gets somewhat buried. The phrasing “Additionally, we further examined differences...” is redundant. Reporting results with specific statistics (β = -0.11, p < 0.0001) in the abstract is good, but the implications of these findings are not adequately synthesized. There’s minimal interpretation beyond stating the association. The phrase “Nurses’ knowledge and attitudes play a crucial role...” is repeated in the abstract and again in the conclusion with near-identical wording, suggesting a lack of synthesis.

Thank you, Reviewer #4 for your construction feedback. We have significantly revised the abstract to remove repetition and ensure the findings are well synthesized and more concise.

Introduction

The introduction excessively relies on citations (e.g., multiple references in one sentence) without critically integrating them. Some references seem included for volume rather than relevance.

Although the issue is important, the introduction lacks a clear conceptual model or hypothesis linking knowledge, attitudes, and caring behavior.

The gap in the literature is stated but not well substantiated. The claim that this is the “first study” to link these constructs lacks sufficient justification or contrast with existing studies.

Terms like “caring behavior” and “attitudes” are introduced without conceptual definitions, relying instead on the instruments (CBI-6, DDPPQ) to carry the burden of meaning.

We appreciate your thoughtful feedback. We have revised the introduction section to strengthen the integration of sources, refine the statement about the gap, and provide definitions of caring behaviors and attitudes based on previous studies. Since a formal hypothesis is not mandated for a descriptive cross-sectional study, our research was exploratory in nature, with specific research questions aimed at investigating demographic and work-related factors that could show differences in nursing knowledge, attitudes, and caring behaviors, as well as how these factors may influence the relationship between these two variables. Future longitudinal and interventional studies with diverse and large samples are needed to explicate whether and how these factors influence caring behaviors.

Methods

The use of convenience and purposive sampling is a significant limitation, but this is not acknowledged until the discussion. This undermines generalizability and introduces bias.

The exclusion of nurse managers and educators is reasonable but not well-justified. Given their role in shaping attitudes and policy, their exclusion may miss important systemic insights.

The explanation of the DDPPQ scoring is confusing. The creation of stratified ranges (e.g., “very low,” “low,” etc.) seems arbitrary without validation or citation.

The methods mention the use of REDCap and a raffle incentive, but there's no reflection on how these may introduce self-selection or response bias.

No justification for the sample size is provided, which weakens confidence in the representativeness and power of statistical analyses.

We have thoroughly revised the method section. Since there are no specific cutoff points for DDPPQ and CBI-6, the research team’s stratification of the instruments was exploratory in nature. Both the independent and dependent measures were used as continuous variables [total scores] when testing the association between the variables. We revised the manuscript to remove mentions of stratification. Instead, we only kept references to the distribution of the measure [range] and internal consistency testing for the studied sample. We also acknowledged the implications of the raffle incentive process via REDcap. Although participation was voluntary, we recognize that motivated participants interested in addiction medicine or research may have been more likely to respond. Incentives may have also increased participation, motivating some participants to complete the survey. To minimize self-selection and response biases, we randomly select raffle winners through an electronic system. Additionally, we ensured that the survey responses were kept separate from the incentive distribution protocol.

Results

The section is filled with statistical comparisons but lacks interpretation. Reporting p-values and CI is appropriate, but there’s minimal explanation of what these mean practically.

The categorization of DDPPQ and CBI-6 scores into subjective ranges (e.g., high, moderate, low) is not grounded in prior literature, reducing replicability.

The volume and complexity of the tables may overwhelm the reader. Many differences are noted as statistically insignificant, suggesting a risk of fishing for significance.

The emphasis on statistical significance ignores potential Type I errors due to multiple comparisons without correction.

Thank you, reviewer 4. In response to your comments regarding the method and results sections, we have revised these sections to strongly highlight all the potential limitations related to convenience and purposive sampling. Additionally, we updated the methods section to include reasons for excluding non-bedside nurses. Since DDPPQ and CBI-6 tools do not have established cut-off points, we conducted a descriptive exploration of the instruments' distribution by creating categorical subunits for each scale. However, we kept the dependent and independent variables as continuous measures. As a result, we have removed the text related to the stratification of both measures from the manuscript. Regarding the tables, several research studies have identified multiple demographic and workplace factors associated with substance use disorder. This study aimed to examine whether some of these factors would show differences in the specific population of patients with opioid use disorder. Only a few showed some differences.

Discussion

The discussion makes strong claims about causality and implications despite the cross-sectional design. Phrases like “nurses’ knowledge and attitudes were the strongest predictors” suggest a causal relationship unsupported by the study design.

Many points are reiterated from earlier sections (e.g., “nurses in community hospitals had less favorable attitudes”), which inflates word count without adding insight.

Although some literature is cited, there’s a lack of critical integration. The findings are not well-situated within broader theoretical or empirical contexts.

Statements about how institutional culture or training might affect outcomes are speculative and unsupported by data from the study.

The limitations section is undercut by generic statements. The gender imbalance and response bias deserve deeper examination.

Thank you, Reviewer 4, for your insightful feedback. We have c

---

## [Decision Letter · Decision Letter 1]

31 Oct 2025

Dear Dr. Essien-Aleksi,

We look forward to receiving your revised manuscript.

Kind regards,

Majed Sulaiman Alamri, PhD

Academic Editor

PLOS ONE

Journal Requirements:

Reviewers' comments:

Reviewer's Responses to Questions

**Comments to the Author**

Reviewer #1: All comments have been addressed

Reviewer #3: All comments have been addressed

Reviewer #5: (No Response)

2. Is the manuscript technically sound, and do the data support the conclusions?

Reviewer #1: Yes

Reviewer #3: Yes

Reviewer #5: Partly

3. Has the statistical analysis been performed appropriately and rigorously?

Reviewer #1: Yes

Reviewer #3: Yes

Reviewer #5: No

4. Have the authors made all data underlying the findings in their manuscript fully available?

Reviewer #1: Yes

Reviewer #3: Yes

Reviewer #5: Yes

5. Is the manuscript presented in an intelligible fashion and written in standard English?

Reviewer #1: Yes

Reviewer #3: Yes

Reviewer #5: Yes

Reviewer #1: (No Response)

Reviewer #3: A power analysis was conducted using G*Power to justify the sample size. The discussion was expanded to address potential response bias, and causal language was carefully avoided throughout the manuscript.

Reviewer #5: The article has been improved across several sections, starting from the title. The topic is relevant and novel. However, its main weakness lies in the methodology section, which leads to the generation of results lacking statistical rigor. The following comments are provided by section:

1. Abstract:

This section can be reduced, as it exceeds 300 words.

2. Introduction:

Improvements in the text are evident after the recent revisions. The section now reads more fluently and has a solid conceptual framework.

3. Method:

o Study Design:

It should be explicitly stated that this is a quantitative study.

o Study Measures:

Caring Behaviors: Instead of stating that Cronbach’s alpha ranges “between 0.89 and 0.89,” it is recommended to indicate that its Cronbach’s alpha coefficient, reported across different studies, is 0.89.

The variable measured by the CBI-6 is discrete. A discrete variable can only take integer, countable values — that is, it results from counting, not measuring.

There are no intermediate values between two consecutive integers.

Such variables are finite and integer-based; therefore, many parametric tests (such as Student’s t-test or ANOVA) are not always appropriate, especially when the number of possible values is small or when the distribution is highly skewed.

For a Student’s t-test to be valid, the following assumptions must be met:

1. The dependent variable is continuous or quasi-continuous (interval or ratio scale).

2. The distribution is approximately normal within each group.

3. Homogeneity of variances across groups.

If the discrete variable does not meet the assumptions of normality or continuity, nonparametric tests should be used instead. These tests do not require normality and are more robust for ordinal or discrete data.

Knowledge and Attitudes (DDPPQ):

This is also a discrete variable, so the same comment applies.

o Covariates:

The decision to categorize continuous variables such as age and years of experience (e.g., 1–5 years, 6–10 years) for bivariate analyses may result in a loss of statistical power and information. If linear regression is used, it is preferable to retain the continuous form of the variable—provided that the assumptions are met.

o Data Analysis:

Unfortunately, the variables were treated as continuous, which is not appropriate since the CBI-6 and DDPPQ scores result from the summation of discrete Likert-type items. Furthermore, a normal distribution of data was assumed, which led to the use of parametric tests such as ANOVA and t-tests.

It is important to note that nonparametric regression models and generalized linear models exist for discrete variables. Moreover, since both CBI and DDPPQ are evaluated through Likert-type scales, it would have been ideal to compute polychoric correlations between the variables.

4. Results:

The tables show highly unbalanced group sizes for key variables. For instance: female = 114, male = 6, other = 1. This large disparity makes group comparisons statistically meaningless. The p-values in such cases have no interpretive validity.

It is recommended that group comparisons be limited to descriptive statistics only. This issue affects most variables and leads to conclusions lacking statistical weight—not only due to unequal group sizes but also because the parametric tests used do not meet the required assumptions.

As previously noted, it is suggested that the authors retain only Research Question No. 2 and compute correlations applicable to the study sample. This approach would be more consistent with a descriptive study design.

5. Discussion and Conclusions:

These sections should be adjusted to align with descriptive results, including correlation analyses.

**Do you want your identity to be public for this peer review?** For information about this choice, including consent withdrawal, please see our Privacy Policy

Reviewer #1: **Yes:** Ola Mousa

Reviewer #3: No

Reviewer #5: **Yes:** Katya Cuadros Carlesi

---

## [Author Response · Author response to Decision Letter 2]

25 Nov 2025

Manuscript Number: PONE-D-25-26604

Title: Association Between Nurses’ Knowledge and Attitudes and Caring Behaviors Toward People with Opioid Use Disorder

Dear Editor and Reviewers,

We sincerely appreciate your additional comments. We have reviewed Reviewer Five's suggestions and made the necessary updates to the manuscript accordingly. Please see our responses below in bold and in the manuscript for more details.

Sincerely,

Inyene Essien-Aleksi, PhD.

Review Comments to the Author

Reviewer #1: (No Response)

Thank you for your feedback.

Reviewer #3: A power analysis was conducted using G*Power to justify the sample size. The discussion was expanded to address potential response bias, and causal language was carefully avoided throughout the manuscript.

Thank you for your comment.

Reviewer #5: The article has been improved across several sections, starting from the title. The topic is relevant and novel. However, its main weakness lies in the methodology section, which leads to the generation of results lacking statistical rigor. The following comments are provided by section:

1. Abstract:

This section can be reduced, as it exceeds 300 words.

We have revised the abstract to limit its length to 300 words. Thank you.

2. Introduction:

Improvements in the text are evident after the recent revisions. The section now reads more fluently and has a solid conceptual framework.

Thank you for your valuable feedback, which greatly enhanced the introduction section of this paper.

3. Method:

o Study Design: It should be explicitly stated that this is a quantitative study.

Thank you for your comment. In the first sentence of the methods section, we now specify that this is a “quantitative cross-sectional study…”

o Study Measures: Caring Behaviors: Instead of stating that Cronbach’s alpha ranges “between 0.89 and 0.89,” it is recommended to indicate that its Cronbach’s alpha coefficient, reported across different studies, is 0.89.

Thanks for your comment. I have updated the sentence to say that “the Cronbach’s α reported across various studies for the CBI-6 was 0.89.”

The variable measured by the CBI-6 is discrete. A discrete variable can only take integer, countable values — that is, it results from counting, not measuring. There are no intermediate values between two consecutive integers. Such variables are finite and integer-based; therefore, many parametric tests (such as Student’s t-test or ANOVA) are not always appropriate, especially when the number of possible values is small or when the distribution is highly skewed.

For a Student’s t-test to be valid, the following assumptions must be met:

1. The dependent variable is continuous or quasi-continuous (interval or ratio scale).

2. The distribution is approximately normal within each group.

3. Homogeneity of variances across groups.

If the discrete variable does not meet the assumptions of normality or continuity, nonparametric tests should be used instead. These tests do not require normality and are more robust for ordinal or discrete data. Knowledge and Attitudes (DDPPQ):

This is also a discrete variable, so the same comment applies.

Thank you for your insightful feedback. The DDPPQ and CBI-6 are both Likert-type scales, which are ordinal at the single-item level, but the composite scores were calculated as the sum of all items in the tools. In nursing research, we generally treat total scores from such scales as continuous (interval-level) data for statistical analysis. For instance, Watson et al. (2007) and Wu et al. (2006) analyzed DDPPQ and CBI-6 scores as continuous variables. Similarly, other studies have followed this approach when examining the psychometric properties and applications of the CBI, including Coulombe et al. (2002) and Edvardsson et al. (2015). These precedents support our decision to treat DDPPQ and CBI-6 scores as continuous rather than categorical or discrete variables.

References:

• Watson, H., Maclaren, W., & Kerr, S. (2007). Staff attitudes towards working with drug users: Development of the Drug Problems Perceptions Questionnaire. Addiction, 102(2), 206–215.

• Wu, Y., Larrabee, J. H., & Putman, H. P. (2006). Caring Behaviors Inventory: A reduction of the 42-item instrument. Nursing Research, 55(1), 18–25.

• Coulombe, K. H., Yeakel, S., Maljanian, R., & Bohannon, R. W. (2002). Caring Behaviors Inventory: Analysis of responses by hospitalized surgical patients. Outcomes Management, 6(3), 138–141.

• Edvardsson, D., Mahoney, A. M., Hardy, J., McGillion, T., McLean, A., Pearce, F., et al. (2015). Psychometric performance of the English language six-item Caring Behaviours Inventory in an acute care context. Journal of Clinical Nursing, 24(17–18), 2538–2544.

Covariates:

The decision to categorize continuous variables such as age and years of experience (e.g., 1–5 years, 6–10 years) for bivariate analyses may result in a loss of statistical power and information. If linear regression is used, it is preferable to retain the continuous form of the variable—provided that the assumptions are met.

Thank you for this insightful comment. It is true that categorizing continuous variables, such as age and years of experience, can reduce statistical power and information. In this study, we used a generalized linear regression model (multiple linear regression) and treated the independent and dependent variables as continuous, while controlling for relevant demographic and work-related factors. The reason we categorized certain covariates, such as age and years of experience, was to examine meaningful group differences and inform future interventions targeting specific groups. For example, whether nurses in certain age groups or with specific years of experience may require more attention or interventions. This approach is also supported by previous studies. For instance, Kravotil et al. (2023) showed that older nurses with more years of nursing experience (stratified into discrete variables) tended to have greater knowledge and more positive attitudes than their younger counterparts (although not consistently across all studies). Stratifying these work-related factors enabled us to balance statistical rigor with practical and empirical implications.

Reference:

• Kratovil A, Schuler MS, Vottero BA, Aryal G. Nurses' self-assessed knowledge, attitudes, and educational needs regarding patients with substance use disorder. American Journal of Nursing. 2023;123(4):26-33.

o Data Analysis:

Unfortunately, the variables were treated as continuous, which is not appropriate since the CBI-6 and DDPPQ scores result from the summation of discrete Likert-type items. Furthermore, a normal distribution of data was assumed, which led to the use of parametric tests such as ANOVA and t-tests.

It is important to note that nonparametric regression models and generalized linear models exist for discrete variables. Moreover, since both CBI and DDPPQ are evaluated through Likert-type scales, it would have been ideal to compute polychoric correlations between the variables.

Thank you for your suggestion to compute polychoric correlations for the DDPPQ and CBI-6 instruments. Although individual items on the DDPPQ and CBI-6 are Likert-type, the composite scores generated by summing all items produced continuous variables. We assessed skewness using the Shapiro–Wilk test and evaluated overall normality for both scales. The knowledge and attitudes scale demonstrated no meaningful skewness and showed a normal distribution, whereas the caring behaviors scale showed mild skewness. Given our large sample size (>30) and the absence of significant outliers in both variables, and the robustness of parametric testing to minor violations of normality assumptions, we concluded that parametric tests such as ANOVA and t-tests were appropriate for our data. We updated the data analysis section to include this note. Finally, Polychoric correlations, rather than Pearson correlations, can certainly reduce the underestimation of correlation that may occur when ordinal data are treated as continuous in Pearson’s r. However, this technique is commonly used in psychometric analysis, structural equation modeling, or factor analysis. Thank you very much for this valuable suggestion.

Reference:

• Field, A. (2024). Discovering statistics using IBM SPSS statistics. Sage Publications.

• Lumley, T., Diehr, P., Emerson, S., & Chen, L. (2002). The importance of the normality assumption in large public health data sets. Annual review of public health, 23(1), 151-169.

4. Results:

The tables show highly unbalanced group sizes for key variables. For instance: female = 114, male = 6, other = 1. This large disparity makes group comparisons statistically meaningless. The p-values in such cases have no interpretive validity.

It is recommended that group comparisons be limited to descriptive statistics only. This issue affects most variables and leads to conclusions lacking statistical weight—not only due to unequal group sizes but also because the parametric tests used do not meet the required assumptions.

As previously noted, it is suggested that the authors retain only Research Question No. 2 and compute correlations applicable to the study sample. This approach would be more consistent with a descriptive study design.

Thank you for your valuable comment. We agree with your point about the limited statistical power to examine group differences due to small sample sizes in some groups. Since some groups remained balanced—such as age, hospital type, work experience, and past OUD experience—we thought it would be helpful to include these results in the paper to give readers additional context. We have addressed this as a major methodological limitation in the Limitations section. Regarding deleting Research question 2: Deleting this research question would significantly change the paper, and we would appreciate the editor’s advice on whether to remove this question. We are happy to make further revisions based on the editor’s suggestions.

5. Discussion and Conclusions:

These sections should be adjusted to align with descriptive results, including correlation analyses.

Thank you for your feedback. We have updated the discussion and conclusion sections to reflect the study results and highlighted the methodology limitations.

---

## [Decision Letter · Decision Letter 2]

9 Dec 2025

We look forward to receiving your revised manuscript.

Kind regards,

Majed Sulaiman Alamri, PhD

Academic Editor

PLOS One

Journal Requirements:

Reviewers' comments:

Reviewer's Responses to Questions

**Comments to the Author**

Reviewer #1: All comments have been addressed

Reviewer #3: All comments have been addressed

Reviewer #5: (No Response)

2. Is the manuscript technically sound, and do the data support the conclusions?

Reviewer #1: Yes

Reviewer #3: Partly

Reviewer #5: Partly

3. Has the statistical analysis been performed appropriately and rigorously?

Reviewer #1: Yes

Reviewer #3: Yes

Reviewer #5: No

4. Have the authors made all data underlying the findings in their manuscript fully available?

Reviewer #1: Yes

Reviewer #3: Yes

Reviewer #5: Yes

5. Is the manuscript presented in an intelligible fashion and written in standard English?

Reviewer #1: Yes

Reviewer #3: Yes

Reviewer #5: Yes

Reviewer #1: The study “Association Between Nurses’ Knowledge and Attitudes and Caring Behaviors Toward People with Opioid Use Disorder” addresses a critical gap in hospital-based care. Its findings demonstrate that nurses’ knowledge levels and attitudes significantly influence the quality of caring behaviors toward patients with opioid use disorder (OUD). This study makes a valuable contribution to nursing science by showing how nurses’ knowledge and attitudes directly shape caring behaviors toward patients with opioid use disorder.

Reviewer #3: The manuscript has undergone substantial improvement and now meets the standards of clarity, coherence, and conceptual validity expected for publication. The authors acknowledge the statistical limitations transparently, and it is ultimately up to the editorial team to determine whether the current analytical approach is acceptable or whether a third review round focused exclusively on methodological refinement is warranted.

Although alternative statistical approaches—such as polychoric correlations or non-parametric models—could have enhanced the methodological rigor, I consider that:

the authors’ justification for maintaining the parametric analysis is methodologically reasonable;

the limitations have been appropriately integrated into the manuscript;

the study’s objective is primarily exploratory and descriptive; and

the findings are interpreted cautiously, without causal language.

In light of these considerations, I believe the manuscript is suitable for acceptance, provided the editorial team is comfortable with the authors’ decision to retain the current parametric analytical framework.

Reviewer #5: Dear Authors,

Although the manuscript has addressed some of the previously identified issues, the quantitative

analysis remains very weak. Of particular concern are the assumptions underlying the data analysis

plan.

The response variables measured by both instruments do not correspond to continuous variables.

These data are obtained through Likert-type scales and, even when multiple items of this kind are

summed, the resulting score does not constitute a numerical variable in the strict sense, nor a discrete

variable in formal statistical terms, and certainly not a continuous one.

With respect to the ANOVA analysis, this term refers to a family of experimental designs that require

the random assignment of study units to factors such as race, age group, nursing role, among others.

None of these conditions are met in the present study. Causal inference would only be feasible if the

formation of groups were truly random. Furthermore, it would be necessary to evaluate potential

interactions among factors, ensure adequate balance across groups (which is also not the case in this

manuscript), and verify that the data originate from an appropriate multivariate probability model—

or a bivariate one if only two variables are analyzed. It is essential to determine whether the data

genuinely conform to a multivariate model. Kim and Timm (2006) propose SAS routines to evaluate

multivariate normality when the data are truly continuous; however, this assumption is not met here.

Likewise, it is not methodologically appropriate to apply Pearson correlation to data that are not

continuous. Under these conditions, assessing normality is also meaningless, given that the variables

are categorical and measured on an ordinal scale. For Likert-type variables with at least five

categories, the appropriate technique for estimating associations is the polychoric correlation.

Nevertheless, reliable estimation of polychoric correlations requires large sample sizes—typically

exceeding 1,000 observations, based on simulation studies—which differs substantially from the 125

units reported in this study, a clearly insufficient number for any robust correlational analysis.

The widely circulated idea that “30 cases are sufficient” is a myth applicable only—if at all—and

very narrowly to univariate variables. The required sample size depends on the variability of the

variables and, in the case of correlations, on the magnitude of the population correlation. When the

absolute value of the correlation between two variables is close to one, very few observations are

needed; however, when correlations are close to zero, required sample sizes clearly exceed 1,000 data

points. For multivariate studies, there are no explicit references supporting the use of multiple

polychoric correlations with small sample sizes.

For all these reasons, and even though the limitations acknowledged by the authors are noted, the

most appropriate course of action would be to present this work as a descriptive, exploratory study.

It may serve as an initial step toward addressing this knowledge gap through future investigations

with larger and randomly selected samples.

Finally, it is important to emphasize that nursing is a science and, as such, must adhere strictly to the

methodological standards that guide the basic sciences. The quality of scientific evidence depends on

the rigor and robustness of the procedures used. The fact that some authors in nursing research have

employed questionable analytical strategies does not legitimize their use. It is the responsibility of the

disciplinary scientific community to safeguard the quality and rigor of published research, thereby

contributing to the strengthening and advancement of the field.

Lastly, I encourage you to review the following authors, whose work may help clarify the appropriate

statistical approach for this study:

• Kiwanuka F, Kopra J, Sak-Dankosky N, Nanyonga RC, Kvist T. Polychoric Correlation with

Ordinal Data in Nursing Research. Nursing Research. 2022;71(6):469–476.

• Kim, K., & Timm, N. (2006). Univariate and Multivariate General Linear Models: Theory

and Applications with SAS. CRC Press.

**Do you want your identity to be public for this peer review?** For information about this choice, including consent withdrawal, please see our Privacy Policy

Reviewer #1: **Yes:** Ola Mousa

Reviewer #3: No

Reviewer #5: **Yes:** Katya Cuadros Carlesi

---

## [Author Response · Author response to Decision Letter 3]

13 Feb 2026

Manuscript Number: PONE-D-25-26604

Title: Association Between Nurses’ Knowledge and Attitudes and Caring Behaviors Toward People with Opioid Use Disorder

Dear Editor and Reviewers,

We thank the reviewers for their careful evaluation of our manuscript and for their constructive and thoughtful feedback. We have revised the manuscript substantially in response to these comments. Below, we provide a detailed, point-by-point response outlining how each concern was addressed. All changes are reflected in the revised manuscript.

Sincerely,

Inyene Essien-Aleksi, PhD.

Review Comments to the Author

Reviewer #1: The study “Association Between Nurses’ Knowledge and Attitudes and Caring Behaviors Toward People with Opioid Use Disorder” addresses a critical gap in hospital-based care. Its findings demonstrate that nurses’ knowledge levels and attitudes significantly influence the quality of caring behaviors toward patients with opioid use disorder (OUD). This study makes a valuable contribution to nursing science by showing how nurses’ knowledge and attitudes directly shape caring behaviors toward patients with opioid use disorder.

We sincerely thank review #1 for the positive assessment of the manuscript.

Reviewer #3: The manuscript has undergone substantial improvement and now meets the standards of clarity, coherence, and conceptual validity expected for publication. The authors acknowledge the statistical limitations transparently, and it is ultimately up to the editorial team to determine whether the current analytical approach is acceptable or whether a third review round focused exclusively on methodological refinement is warranted.

Although alternative statistical approaches—such as polychoric correlations or non-parametric models—could have enhanced the methodological rigor, I consider that: the authors’ justification for maintaining the parametric analysis is methodologically reasonable; the limitations have been appropriately integrated into the manuscript; the study’s objective is primarily exploratory and descriptive, and the findings are interpreted cautiously, without causal language.

In light of these considerations, I believe the manuscript is suitable for acceptance, provided the editorial team is comfortable with the authors’ decision to retain the current parametric analytical framework.

We thank Reviewer #3 for their thoughtful assessment and for recognizing the improvements made to the manuscript. In response to both Reviewer #3 and Reviewer #5, we further strengthened the manuscript’s alignment with an explicitly descriptive and exploratory analytic framework. Specifically, we removed all parametric analyses and reframed the statistical approach to rely exclusively on non-parametric and ordinal-appropriate methods. We believe these revisions further reinforce the transparency and methodological coherence noted by Reviewer #3 and reduce any remaining concerns regarding analytical assumptions.

Reviewer #5: Dear Authors,

Although the manuscript has addressed some of the previously identified issues, the quantitative

analysis remains very weak. Of particular concern are the assumptions underlying the data analysis

plan.

The response variables measured by both instruments do not correspond to continuous variables.

These data are obtained through Likert-type scales and, even when multiple items of this kind are

summed, the resulting score does not constitute a numerical variable in the strict sense, nor a discrete

variable in formal statistical terms, and certainly not a continuous one.

With respect to the ANOVA analysis, this term refers to a family of experimental designs that require

the random assignment of study units to factors such as race, age group, nursing role, among others.

None of these conditions are met in the present study. Causal inference would only be feasible if the

formation of groups were truly random. Furthermore, it would be necessary to evaluate potential

interactions among factors, ensure adequate balance across groups (which is also not the case in this

manuscript), and verify that the data originate from an appropriate multivariate probability model—

or a bivariate one if only two variables are analyzed. It is essential to determine whether the data

genuinely conform to a multivariate model. Kim and Timm (2006) propose SAS routines to evaluate

multivariate normality when the data are truly continuous; however, this assumption is not met here.

Likewise, it is not methodologically appropriate to apply Pearson correlation to data that are not

continuous. Under these conditions, assessing normality is also meaningless, given that the variables

are categorical and measured on an ordinal scale. For Likert-type variables with at least five

categories, the appropriate technique for estimating associations is the polychoric correlation.

Nevertheless, reliable estimation of polychoric correlations requires large sample sizes—typically

exceeding 1,000 observations, based on simulation studies—which differs substantially from the 125

units reported in this study, a clearly insufficient number for any robust correlational analysis.

The widely circulated idea that “30 cases are sufficient” is a myth applicable only—if at all—and

very narrowly to univariate variables. The required sample size depends on the variability of the

variables and, in the case of correlations, on the magnitude of the population correlation. When the

absolute value of the correlation between two variables is close to one, very few observations are

needed; however, when correlations are close to zero, required sample sizes clearly exceed 1,000 data

points. For multivariate studies, there are no explicit references supporting the use of multiple

polychoric correlations with small sample sizes.

For all these reasons, and even though the limitations acknowledged by the authors are noted, the

most appropriate course of action would be to present this work as a descriptive, exploratory study.

It may serve as an initial step toward addressing this knowledge gap through future investigations

with larger and randomly selected samples.

Finally, it is important to emphasize that nursing is a science and, as such, must adhere strictly to the

methodological standards that guide the basic sciences. The quality of scientific evidence depends on

the rigor and robustness of the procedures used. The fact that some authors in nursing research have

employed questionable analytical strategies does not legitimize their use. It is the responsibility of the

disciplinary scientific community to safeguard the quality and rigor of published research, thereby

contributing to the strengthening and advancement of the field.

Lastly, I encourage you to review the following authors, whose work may help clarify the appropriate

statistical approach for this study:

• Kiwanuka F, Kopra J, Sak-Dankosky N, Nanyonga RC, Kvist T. Polychoric Correlation with

Ordinal Data in Nursing Research. Nursing Research. 2022;71(6):469–476.

• Kim, K., & Timm, N. (2006). Univariate and Multivariate General Linear Models: Theory

and Applications with SAS. CRC Press.

We thank Reviewer #5 for the detailed and rigorous critique. We carefully considered all concerns raised, consulted with a senior statistician at our institution, and fully revised the data analysis plan and manuscript to address these issues. The revised manuscript now explicitly adopts an ordinal, non-parametric, and exploratory analytical framework, consistent with the reviewer’s recommendations. The key revisions are summarized below:

1. Removal of parametric analyses

All parametric tests (including ANOVA, Pearson correlation, and assumptions of normality or interval-level measurement) have been removed. The manuscript no longer treats DDPPQ or CBI-6 scores as continuous variables.

2. Explicit treatment of DDPPQ and CBI-6 as ordinal composite measures

We now clearly state that both instruments are composed of Likert-type items and that summed scores were treated as ordinal composite scores, not continuous variables. Measures of central tendency are reported using medians and interquartile ranges, and no claims are made regarding interval-level measurement.

3. Use of appropriate non-parametric methods

Group differences are examined using Mann–Whitney U tests and Kruskal–Wallis tests, with post hoc comparisons conducted using the Dwass–Steel–Critchlow–Fligner method where appropriate (via SAS). Associations between nurses’ knowledge/attitudes and caring behaviors are assessed using Spearman’s rank correlation, with bootstrapped confidence intervals to improve robustness.

4. Adoption of median (quantile) regression

To examine adjusted associations, we replaced linear regression with median (τ = 0.50) quantile regression, which does not assume normality or interval-level data and is appropriate for skewed ordinal outcomes. This approach was selected due to the observed ceiling effects in CBI-6 scores and the absence of validated clinical cut-off points.

5. Clear framing as descriptive and exploratory

Throughout the manuscript, we explicitly describe the study as descriptive and exploratory. All causal language has been removed, and findings are consistently interpreted as associational rather than inferential or causal.

6. Transparency regarding subgroup sizes and data limitations

We now clearly document the merging of small subgroups, the exclusion of categories with insufficient cell sizes, and the rationale for these decisions. These limitations are discussed in detail in the Methods and Limitations sections.

7. Expanded limitations and future directions

The revised Discussion and Limitations sections explicitly acknowledge the constraints of ordinal measurement, non-random sampling, subgroup imbalance, and sample size. We emphasize that the findings serve as an initial step to inform future studies with larger, randomly selected samples and more advanced modeling approaches.

We believe these revisions directly address Reviewer #5’s methodological concerns and align the manuscript with accepted statistical standards for ordinal data analysis in nursing and health sciences research. Importantly, the study’s purpose is now clearly positioned as generating preliminary, hypothesis-informing evidence rather than definitive inference. Once again, we appreciate the reviewers’ careful evaluations and believe the revised manuscript is substantially strengthened as a result of the feedback.

Thank you very much.

Best,

Inyene Essien-Aleksi

---

## [Decision Letter · Decision Letter 3]

2 Mar 2026

Association Between Nurses’ Knowledge and Attitudes and Caring Behaviors Toward People with Opioid Use Disorder

PONE-D-25-26604R3

Dear Dr. Inyene,

We’re pleased to inform you that your manuscript has been judged scientifically suitable for publication and will be formally accepted for publication once it meets all outstanding technical requirements.

Kind regards,

Majed Sulaiman Alamri, PhD

Academic Editor

PLOS One

Additional Editor Comments (optional):

Reviewers' comments:

Reviewer's Responses to Questions

**Comments to the Author**

Reviewer #1: All comments have been addressed

Reviewer #3: All comments have been addressed

Reviewer #5: All comments have been addressed

2. Is the manuscript technically sound, and do the data support the conclusions?

Reviewer #1: Yes

Reviewer #3: Yes

Reviewer #5: Yes

3. Has the statistical analysis been performed appropriately and rigorously?

Reviewer #1: Yes

Reviewer #3: Yes

Reviewer #5: Yes

4. Have the authors made all data underlying the findings in their manuscript fully available?

Reviewer #1: Yes

Reviewer #3: Yes

Reviewer #5: Yes

5. Is the manuscript presented in an intelligible fashion and written in standard English?

Reviewer #1: Yes

Reviewer #3: Yes

Reviewer #5: Yes

Reviewer #1: The study was complete in structure and addressed all required elements, with the researchers responding to reviewers’ comments in a clear and academic manner.

No more comments.

Reviewer #3: The authors have substantially addressed the reviewers’ methodological concerns. In the revised manuscript, they removed all parametric analyses (e.g., ANOVA, Pearson correlation, normality testing) and no longer treat DDPPQ/CBI-6 scores as continuous. Instead, they explicitly frame these summed Likert measures as ordinal composite outcomes, report medians/IQRs, and use appropriate nonparametric tests (Mann–Whitney U, Kruskal–Wallis with post hoc procedures). Associations are assessed with Spearman correlations (with bootstrap CIs), and adjusted analyses use median (quantile) regression. Finally, the study is clearly positioned as descriptive/exploratory, with causal language removed and limitations expanded.

Reviewer #5: Dear authors, I appreciate the positive reception of my previous review. The changes you have made have substantially improved the manuscript, both methodologically and in the analysis of the results, which in turn has enriched the discussion section.

My only suggestion is that you do not mention the statistical technique used to analyze the data again in the results section, as this information is already included in the methodology section.

Under these conditions, I believe the article is ready for publication. I hope that the results of this research will contribute to the nursing care of patients with Opioid Use Disorder.

**Do you want your identity to be public for this peer review?** For information about this choice, including consent withdrawal, please see our Privacy Policy

Reviewer #1: **Yes:** Ola Mousa

Reviewer #3: No

Reviewer #5: **Yes:** Katya Cuadros Carlesi

---

## [Editor Report · Acceptance letter]

PONE-D-25-26604R3

PLOS One

Dear Dr. Essien-Aleksi,

I'm pleased to inform you that your manuscript has been deemed suitable for publication in PLOS One. Congratulations! Your manuscript is now being handed over to our production team.

Kind regards,

on behalf of

Prof. Majed Sulaiman Alamri

Academic Editor

PLOS One